# Spatio-Temporal Classification Framework for Mapping Woody Vegetation from Multi-Temporal Sentinel-2 Imagery

**Jovan Kovačević [1,*](#), Željko Cvijetinović [1](#), Dmitar Lakušić [2], Nevena Kuzmanović [2](#), Jasmina Šinžar-Sekulić [2], Momir Mitrović [3], Nikola Stančić [1](#), Nenad Brodić [1] and Dragan Mihajlović [1]**

1   Faculty of Civil Engineering, University of Belgrade, Bulevar kralja Aleksandra 73, 11000 Belgrade, Serbia; zeljkoc@grf.bg.ac.rs (Ž.C.); nstancic@grf.bg.ac.rs (N.S.); nbrodic@grf.bg.ac.rs (N.B.); draganm@grf.bg.ac.rs (D.M.)
2   Faculty of Biology, University of Belgrade, Institute of Botany and Botanical Garden "Jevremovac", Takovska 43, 11000 Belgrade, Serbia; dlakusic@bio.bg.ac.rs (D.L.); nkuzmanovic@bio.bg.ac.rs (N.K.); jsekulic@bio.bg.ac.rs (J.Š.-S.)
3   MapSoft d.o.o, Ustanička 64/7, 11000 Belgrade, Serbia; momir@mapsoft.rs
*   Correspondence: jkovacevic@grf.bg.ac.rs

**Abstract:** The inventory of woody vegetation is of great importance for good forest management. Advancements of remote sensing techniques have provided excellent tools for such purposes, reducing the required amount of time and labor, yet with high accuracy and the information richness. Sentinel-2 is one of the relatively new satellite missions, whose 13 spectral bands and short revisit time proved to be very useful when it comes to forest monitoring. In this study, the novel spatio-temporal classification framework for mapping woody vegetation from Sentinel-2 multitemporal data has been proposed. The used framework is based on the probability random forest classification, where temporal information is explicitly defined in the model. Because of this, several predictions are made for each pixel of the study area, which allow for specific spatio-temporal aggregation to be performed. The proposed methodology has been successfully applied for mapping eight potential forest and shrubby vegetation types over the study area of Serbia. Several spatio-temporal aggregation approaches have been tested, divided into two main groups: pixel-based and neighborhood-based. The validation metrics show that determining the most common vegetation type classes in the neighborhood of 5 × 5 pixels provides the best results. The overall accuracy and kappa coefficient obtained from five-fold cross validation of the results are 82.97% and 0.75, respectively. The corresponding producer's accuracies range from 36.74% to 97.99% and user's accuracies range from 46.31% to 98.43%. The proposed methodology proved to be applicable for mapping woody vegetation in Serbia and shows a potential to be implemented in other areas as well. Further testing is necessary to confirm such assumptions.

**Keywords:** classification; Sentinel-2; woody vegetation; probability random forest; forest inventory; Serbia

## 1. Introduction

Forests represent the most dominant land ecosystem, which covers about 31 percent of the Earth's total land area [1]. The forests can be sorted into number of different types, commonly defined using information about the composition of species, productivity or crown closure [2]. Mapping woody vegetation is of crucial importance to properly understand forest ecosystems. Providing accurate and cost-efficient information about the forest is critical for proper forest management, ecosystem

preservation and climate change mitigation [3–5]. Traditional field survey methods provide highly accurate forest inventory, yet these methods are highly time-consuming and a lot of manual labor is required. The advancements of the remote sensing techniques try to fill such gaps by reducing the required amount of time and labor, yet with the preservation of the accuracy and the information richness [6–8].

Optical sensors are widely used to identify woody vegetation, although this type of forest inventory remains a challenge. In addition to common problems that affect all optical remote sensing, such as weather conditions, illumination and cloud cover, there are also some issues that particularly affect forest inventory. The spectral separability between the tree species can be influenced by forest maturity and tree density in combination with leaf biochemical properties and canopy structure. Thus, the spectral variability within species may be higher than the variability between species [7]. Furthermore, a continuous transition between vegetation types is characteristic for some tree species, making it difficult to spectrally differentiate such changes [9]. Sentinel-2 is a relatively new satellite mission equipped with MultiSpectral Instrument (MSI). Sentinel-2 collects data in 13 spectral bands, with spatial resolution from 10 m to 60 m, depending on the spectral channel. With constellation of two twin-satellites, Sentinel-2 has a very short revisit time, ideally providing land cover observations every 5 days under cloud-free conditions [10]. Such characteristics proved to be very useful when it comes to vegetation monitoring, with many studies specifically aiming at forest monitoring and inventory [11–13]. Multiple studies proved that the use of time-series improves the tree species identification compared to using only single date satellite images [12,14,15]. This way, the species phenology can be exploited, but it also induces some new problems, such as increasing the dimensionality of the data and the requirements for more complex algorithms, and longer processing times [16,17].

Machine learning (ML) techniques proved to be very useful when it comes to mapping woody vegetation from remotely sensed data [6,18]. Random forest (RF) is one of the vastly popular ML algorithms, which yields very high accuracy of results even with high-dimensional, noisy, and multi-source data [19]. Unfortunately, RF does not account for spatial and temporal characteristics per se, and therefore such information needs to be taken into account in some other way. A number of approaches have been proposed in order to adapt to the spatio-temporal aspects in the RF model. Temporal aspects are commonly accounted for implicitly by defining a separate set of variables for each timestamp [6,7], or explicitly by including such information as one or more variables (day of year, cumulative day of year, etc.) [20]. Spatial aspects can also be taken into account in the same way by including coordinates as variables in the model [20,21]. There are also more complicated solutions such as the one proposed by Sekulić et al., where the spatially and temporally nearest observations are included explicitly as variables [22]. In the case of land cover and land use classification problems, the neighborhood of the pixels proved to be very important. Aonpong et al. improved land cover prediction by involving the neighboring pixels [23]. Samardžić-Petrović et al. used the most common class in the Moore neighborhood for predicting the short term land-use changes, which proved to be one of the most informative variables in the created model [24]. To the best of the authors' knowledge, the applicability of neighborhood-based approaches has not yet been tested for woody vegetation mapping.

The more common approach to defining a woody vegetation prediction model is to omit temporal variables from the model and instead include temporal information implicitly by creating separate sets of variables of all spectral bands for each timestamp [6,7]. This means that the number of variables in the model grows as there are more timestamps available. For too many timestamps, this can lead to computational and performance issues. However, the bigger problem is that this approach, in most cases, requires that each part of the study area has observations available for all timestamps used. Therefore, especially in the case of using optical data, the timestamps with clouds must be avoided. As the study area increases, it becomes more difficult to obtain cloud-free data with similar timestamps. For some large areas, it is even impossible to have completely cloud-free data. There are several solutions for this problem, each with its pros and cons. One way is to decrease the temporal

resolution and use monthly or yearly timestamps. Such generalization introduces additional errors in the model and can miss the fine phenological vegetation patterns. There have also been various attempts to build methods specifically to fill missing data gaps due to cloud cover using different kinds of techniques [25–27]. These approaches add an extra layer of complexity, reduce spatial resolution and/or are not good enough when the cloud gaps are too large.

In this study, the new spatio-temporal classification framework has been proposed for mapping woody vegetation from multi-temporal Sentinel-2 imagery. The method exploits the benefits of explicitly defining temporal variables in the model and the use of pixels' neighborhoods to determine the most probable class of woody vegetation. This was done by using each pixel's class probability prediction obtained from the RF model together with its neighboring pixels' prediction to improve output class estimation. The framework has been tested for mapping forests over Serbia, using a classification scheme with eight potential forest and shrubby vegetation types.

## 2. Materials and Methods

### 2.1. Study Area

The study area is the complete area of the Republic of Serbia, which covers 88,361 km$^2$ (Figure 1). The northern part of the country belongs to the Pannonian Plain, where elevation in most part is less than 200 m. The central parts of Serbia are dominated by hills, which turn into mountains in the south, with more than 30 mountain peaks higher than 2000 m. The humid continental climate is present throughout most of the country, whereas the alpine climate can be found in higher mountains. The average amount of precipitation is 896 mm.

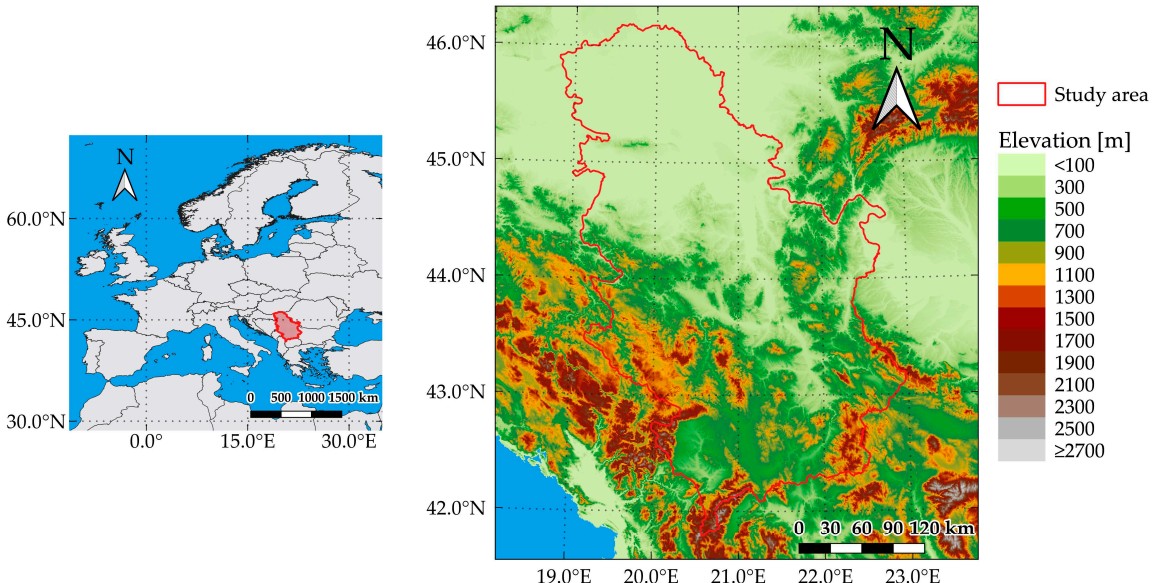

**Figure 1.** The location of study area—Republic of Serbia.

The land cover is in accordance with the relief of Serbia, where agricultural land covers most of the Pannonian Plain and the plains around larger rivers. The rest of the area is mainly covered by forests and grasslands. Wetlands and barren land area are also present in certain parts of the country. Forests cover more than a third of the area, with species of oak, beech, pines and firs being the most common [28]. According to Bohn et al. [29] the most common forest types in Serbia are thermophilous mixed deciduous broadleaved forests (G.1, G.2, G.3—forest dominated by *Quercus cerris*, *Q. frainetto*, *Q. petraea*, *Q. pubescens*, *Carpinus orentalis*, *Ostrya carpinifolia*) and mesophytic broadleaved deciduous (F.3, F5—forest dominated by *Fagus sylvatica*, *F. moesiaca*, *Carpinus betulus*). In addition, significant areas are covered with mesophytic coniferous forests (D.4, D.5, D.6—forest dominated by *Abies alba*,

*Picea abies, P. omorika, Pinus sylvestris, Pinus peuce*); xerophytic coniferous forests, coniferous woodland and scrub (K.1—forest dominated by *P. nigra* agg., *P. heldreichii*); forest steppes (L.1, L.2—forest dominated by *Quercus pubescens, Q. robur, Q. pedunculiflora, Acer tataricum*) and softwood alluvial forests (U.5—forest dominated by *Populus nigra, P. alba, Salix alba*).

### 2.2. Data

#### 2.2.1. Sentinel-2 Imagery

The Sentinel-2 mission is a part of the Copernicus Programme developed by the European Space Agency (ESA) [10]. It is equipped with a MultiSpectral Scanner that has 13 spectral bands—4 bands with 10 m spatial resolution, 6 with 20 m spatial resolution and the remaining 3 bands with 60 m spatial resolution. In this research, Level-2A Sentinel-2 products have been used. They represent orthorectified Bottom-Of-Atmosphere (BOA) reflectance [30]. The product was disseminated as $100 \times 100$ km granules, with bands resampled to 10 m, 20 m and 60 m spatial resolution with a corresponding Scene Classification map (SCL). The SCL map differentiates several basic classes (cloud, cloud shadows, vegetation, soils/deserts, water, snow, etc.) [30]. All of the available 60 m resampled bands (B01, B02, B03, B04, B05, B06, B07, B08a, B09, B11 and B12) and 60 m SCL maps have been used in this research.

A total of 88 Sentinel-2 granules from the year of 2019 have been obtained from Copernicus Open Access Hub (https://scihub.copernicus.eu/). Only the year 2019 was chosen in order to reduce the ambiguities caused by forest changes, which occurred over multiple years. At the same time, this made the woody vegetation map very recent and applicable to other ongoing studies/research over the study area. Granules were selected to have less than 20% cloud cover and so that each part of the study area was covered by at least one granule in each of the time intervals: (1) June–July, (2) August, (3) September and (4) October. The months before June and after October were severely influenced by cloud contamination and snow cover at higher altitudes, which made them not particularly useful for the study area of interest. In addition, these four periods were selected in order to take advantage of phenological patterns for differentiating the types of woody vegetation. All types of woody vegetation were in full vegetation in June and their vegetation type specifically reduced as the colder period approached. The model aimed to capture these patterns to detect different types of woody vegetation. The timestamps and spatial coverage of the acquired granules can be seen in Figure 2.

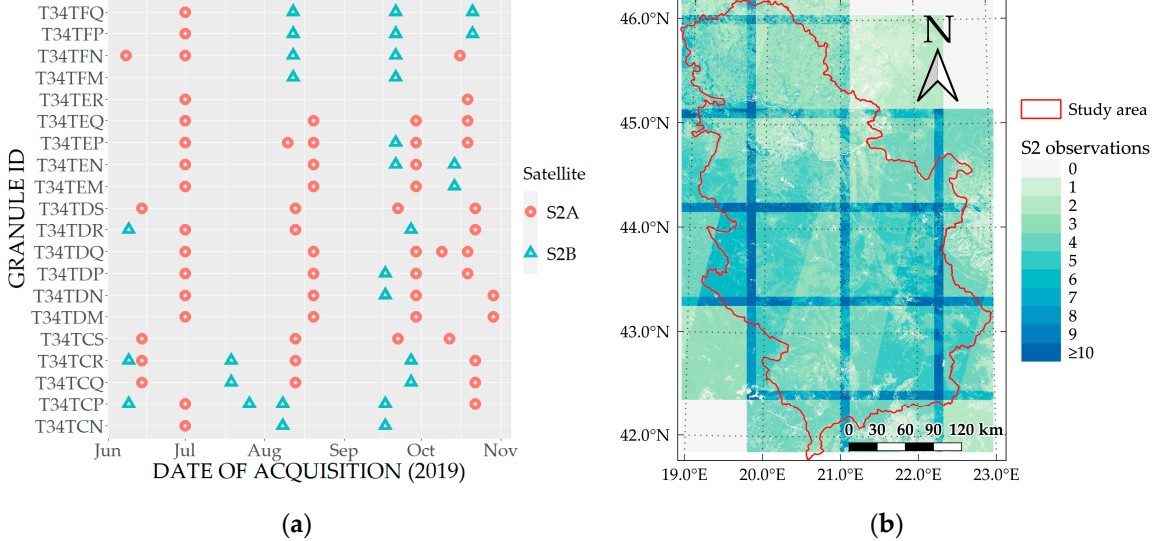

(a)　　　　　　　　　　　　　　　　　　　　　　　　　　　　(b)

**Figure 2.** The used Sentinel-2 granules over the study area: (**a**) Timestamps the granules; (**b**) Number of Sentinel-2 observations available over the study area.

2.2.2. Woody Vegetation Polygons

The "Database on the distribution of potentially endangered species and habitats of Serbia" [31], created through extensive field campaign and over multiple seasons, was used as a base source for the delineation of woody vegetation types. The locations of species were extracted from the database and used to create buffers with a radius of 500 m. These buffers were used as a "base map" to help in the manual vectorization of woody vegetation polygons using Google Earth software and high-resolution satellite imagery. The woody vegetation polygons were created using sufficiently large zoom to reduce vectorization errors and to be as homogenous as possible. The historical imagery (representing different seasons) was also used where necessary to solve ambiguities. The 8 potential forest and shrubby vegetation types classes have been chosen for this research, as they represent the main woody vegetation classes present in the study area. The resulting dataset consists of 549 polygons representing 7 forest and 1 shrubby vegetation classes. The definition of classes and their individual representation, expressed as number of polygons and covered area, are given in Table 1. For an easier interpretation of the results and discussion, informal phytosociological names, without a syntaxonomic meaning, were assigned to each class. The corresponding European Nature Information System (EUNIS) [32] and SrbHab code [33] was determined for each class (Appendix A).

**Table 1.** The definition of classification scheme and the number of vectorized training polygons for their representation.

| Class ID | Vegetation Type | Informal Phytosociological Name | EUNIS Code | SrbHab Code | Number of Polygons |
|---|---|---|---|---|---|
| 1 | Thermophytic broadleaved deciduous (*Carpinus orientalis–Ostrya carpinifolia*) forests | Ostryo-Carpinion | G1.7C | A2.9 | 40 |
| 2 | Thermophytic broadleaved deciduous (Quercus) forests | Quercion frainetto | G1.76 | A2.1 | 63 |
| 3 | Thermo–Mesophytic broadleaved deciduous (Quercus) forests | Quercion petraea-cerris | G1.76 | A2.5 | 107 |
| 4 | Mesophytic and hygromesophytic broadleaved deciduous (Quercus) forests | Quercion roboris | G1.2 and G1.A | A1.3 and A1.4 | 116 |
| 5 | Mesophytic broadleaved deciduous (Fagus) forests | Fagion | G1.6 | A3.2 | 93 |
| 6 | Xerophytic relic coniferous (Pinus) forests | Pinion nigrae | G3.5 | A5.1 | 55 |
| 7 | Psychrophytic boreomontane coniferous (Picea) forests | Vaccinio-Piceion | G3.1 | A6.1 | 50 |
| 8 | Psychrophytic subalpine coniferous krummholz (Pinus) scrub | Pinion mugo | F2.4 | B6.1 | 25 |

It was not possible to create the woody vegetation polygons with a specific spatial distribution, nor with a distribution that would respect the flat and mountainous areas in the study area. This is because the northern flat part of the study area has only a small portion of forest cover with only certain types of woody vegetation. Thus, some types of woody vegetation are found only in certain parts of the study area. In addition, the locations for woody vegetation polygons were limited by data available in the "Database on the distribution of potentially endangered species and habitats of Serbia". Although a number of clusters of polygons are present, there is no class that is represented by a single polygon cluster (Figure 3).

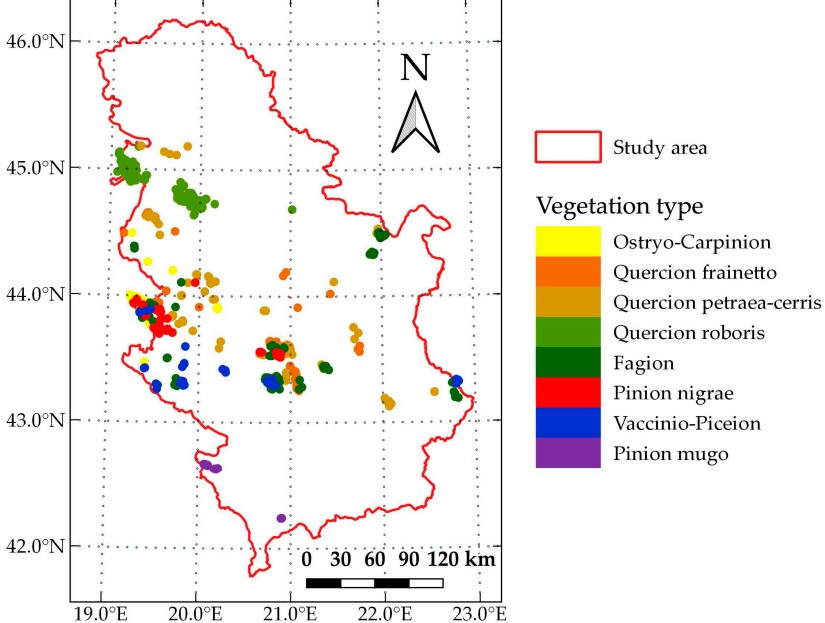

**Figure 3.** The overview of created woody vegetation polygons' locations over the study area.

### 2.2.3. Forest Mask

The Copernicus Land Monitoring Service—High Resolution Layer Forest product has been used in order to create a forest mask. This was necessary in order to remove non-forest areas from the study area. High Resolution Layer Forest consists of multiple products, which are created from the multitemporal Sentinel-2A and Landsat 8 data [34]. These products are: (1) Tree Cover Density, (2) Dominant Leaf Type, (3) Forest Type and (4) Forest Additional Support Layer. All of these products are available over the complete area of Europe in 20 m and 100 m spatial resolutions for the years 2012 and 2015.

From several available products, the Forest Type product and the Forest Additional Support Layer have been acquired, both in the spatial resolution of 20 m and for year 2015. All types of forest areas available in the Forest Type product have been reclassified into a binary mask. Trees in urban or agricultural contexts have been excluded from such a mask. This has been done by using information on such areas that is available in Forest Additional Support Layer products.

### 2.3. Synthetic Minority Oversampling TEchnique (SMOTE)

The unbalanced classes in the dataset, representing a situation where one class in the training set dominates the others, is a common classification problem that can severely influence the performance of the classification model. The Synthetic Minority Oversampling TEchnique (SMOTE) is a popular approach for dealing with unbalanced datasets [35]. This method works by artificially creating additional examples of the minority class using the nearest neighbors' examples. Some SMOTE implementations simultaneously undersample the majority classes, which produces a more balanced dataset in the end and reduces the amount of data. The SMOTE algorithm implemented in the DMwR R package has been used in this research for balancing the extracted dataset of woody vegetation classes [35,36].

### 2.4. Probability Random Forest

Random forest (RF) is a non-parametric supervised machine learning technique proposed by Breiman [37] that works for both classification and regression problems. The main idea behind RF is to build a set of decision trees in the training phase. Each tree is created by bootstrap sampling part of the input data while the rest of the input data is used to evaluate performance and adjust the building

process of the trees. When the forest of decision trees is created, the final prediction is determined as mean prediction of all created decision trees in case of a regression, or as a most common predicted class in classification [37].

Probability random forest is an enhancement of the random forest method that provides consistent probability estimation [38]. In case of binary classification, this is done by using the proportion of "1s" in the output of the trained classification forest. Such an approach is generalized for multiclass classification problems as well. In this research, the probability random forest method implemented in the Ranger R package has been used [36,39].

The RF method was chosen over other ML techniques because of its simple implementation, robustness and interpretability of the results. Additionally, it has been proven to perform well for tree species and forest vegetation detection in similar studies [6,40,41]. Unlike the standard random forest classifier where the output is a plain class label, probability RF also provides the probabilities for every class to be voted. These probability estimates allow for more realistic representation of the ambiguity of phenomena and are suitable to perform the spatio-temporal aggregation.

## 3. Methodology

There are several processing steps of the proposed spatio-temporal framework for woody vegetation classification (Figure 4). First, the training polygons are overlaid with available Sentinel-2 granules in order to extract spectral information and to calculate the Normalized Difference Vegetation Index (NDVI) value for each observation. This was done for each pixel over the study area. The classes of the extracted dataset are then balanced using the SMOTE algorithm. Next, the probability random forest model is grown and used to predict the class probabilities for each pixel. The spatio-temporal aggregation was done in the final step in order to determine the most probable woody vegetation classes over the study area. The obtained results were assessed using five-fold and 10-fold leave-location out cross-validation (LLOCV), by calculating the parameters from the confusion matrix. These calculated parameters are user's accuracy, producer's accuracy, overall accuracy and kappa coefficient of agreement [42]. All these steps are further explained in the following sections.

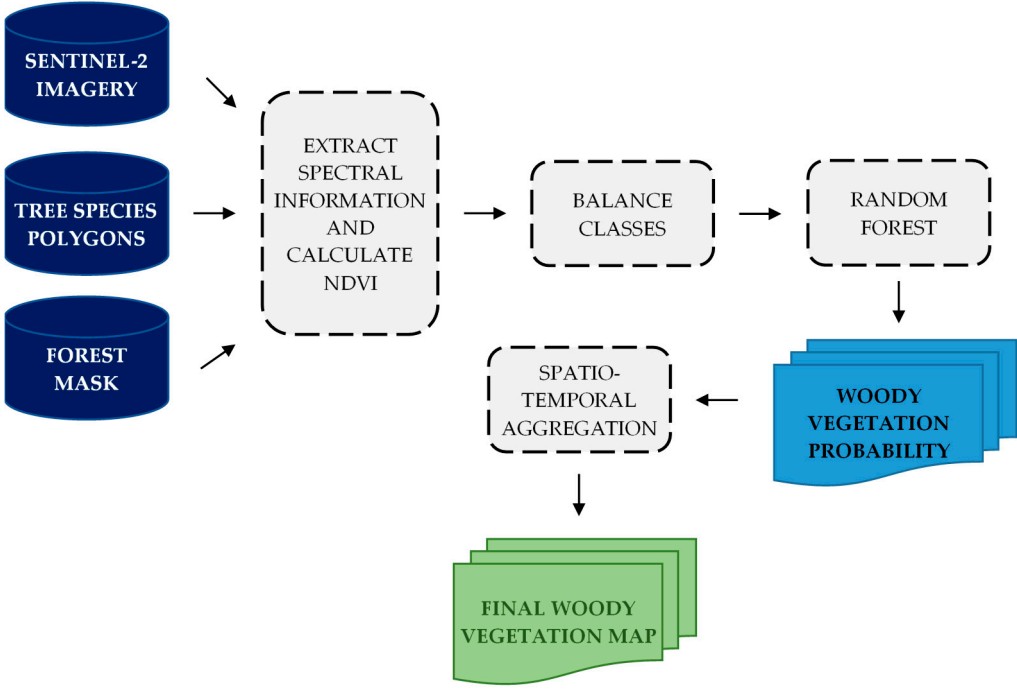

**Figure 4.** The flowchart diagram of the proposed spatio-temporal classification framework.

### 3.1. Extraction of Spectral Information, NDVI Calculation and Balancing of Classes

The spectral information was extracted, overlaying the woody vegetation polygons with the acquired Sentinel-2 granules. The extraction was done per granule, where all pixels within the training polygons were considered as independent observations. The spectral information of all resampled 60 m bands was extracted in this way for all pixels that were labeled as vegetation class on the Sentinel-2 scene classification map. Since the vegetation class did not represent only forest areas, the forest mask was additionally applied to filter out the non-forest pixels. Although there were many different vegetation indices (VI), the Normalized Difference Vegetation Index (NDVI) was chosen as it is one of the most widely used vegetation indices. Other spectral indices might have improved the model performance, but they would have required considerations regarding the selection criteria and their effect on the methodology performance. However, the conclusion was that this would extend the paper significantly, so it was decided to leave this topic for future studies. Therefore, only the corresponding NDVI value has been calculated for each observation using the following formula:

$$\text{NDVI} = \frac{\text{B08a} - \text{B04}}{\text{B08a} + \text{B04}} \tag{1}$$

where B08a and B04 represent the narrow NIR and Red Sentinel-2 spectral bands, respectively. Band B08a was chosen instead of band B08 because it came already resampled in 60 m spatial resolution in the Level-2A Sentinel-2 product distribution. Additionally, the narrower spectral width of band B08a should help to better distinguish woody vegetation classes compared to the use of spectrally wider band B08.

In order to deal with the unbalanced classes, SMOTE has been applied. The SMOTE has been used in such a way that no under sampling was performed at all and that the balancing was only done by creating the examples of the minority classes. This was repeated until there were no more than 10% differences between the amounts of class examples.

### 3.2. Growing Probability Forest and Determination of the Probability for Each Class

The probability random forest for woody vegetation classification has been grown using the previously generated balanced dataset. In order to avoid data leakage in the time series data, the RF model was defined for each S2 observation (not per pixel) with the following formula:

$$\text{WOODY}_{\text{VEGETATION}} = \text{day} + \text{month} + \text{B01} + \text{B02} + \text{B03} + \text{B04} + \text{B05} + \text{B06} + \text{B07} + \\ \text{B08}a + \text{B09} + \text{B11} + \text{B12} + \text{NDVI} \tag{2}$$

where day and month represent the timestamp of the granule that each observation belongs to, $B_{xx}$ is the corresponding Sentinel-2 spectral band and NDVI is the previously calculated vegetation index. The same reasons for choosing band B08a over band B08 previously stated for NDVI calculation also apply to the RF model. Two variants of defining timestamps in the model were tested. The first one assumed the use of two variables (day and month) and the second one only one variable (Day of Year—DOY) in order to make model more robust. The test indicated that the first variant provides slightly better model performances, so it was selected as the final model in the study. Currently, the authors cannot provide a plausible explanation for this behavior of the model.

### 3.3. Spatio-Temporal Aggregation

Since each part of the study area was covered with multiple Sentinel-2 granules, there were several class probability predictions available for each pixel of the study area. Such multiple predictions per pixel had different timestamps and allowed for temporal aggregation. Spatial aspects were also taken into account, since the corresponding pixels and their neighborhoods had been used. All these aspects were taken into account in the last step and the aggregated woody vegetation class predictions were produced for every pixel of the study area.

Several types of spatio-temporal aggregations have been tested. They were separated into two main groups: pixel-based and neighborhood-based. The pixel-based method means that the aggregation function is applied using all available class probability predictions per pixel of the study area. The neighborhood-based aggregation uses the class probability predictions of the surrounding pixels together with predictions for the central pixel. In this research, the neighborhood-based method was implemented using focal raster operation with $3 \times 3$ and $5 \times 5$ kernels, for which the appropriate aggregation function was used. Several aggregation functions were tested, where the final woody vegetation class was determined using some of the following rules:

1. Most common class (MC);
2. Class with the highest probability—simple mean (SM);
3. Class with the highest probability—geometric mean (GM).

The most common class rule means that each prediction from the set of multiple predictions casts a vote to the class with highest probability. The final class is then set to the class that has the highest number of votes. The other two approaches are based on averaging expert predictions [43]. In these cases, each probability prediction was considered as a single expert prediction, which were then averaged by calculating the simple mean or geometric mean. No transformation on predictions was performed and no weights were used in the averaging. The final class was then set to the class that had the highest averaged probability.

### 3.4. Validation

The performance of the final woody vegetation class predictions was validated using K-fold leave-location out cross-validation (LLOCV). Leave-location-out was implemented in a way that the entire woody vegetation polygon (with all corresponding observations) was assigned to the same fold. The complete dataset was therefore randomly split into K groups, or folds, of similar size. The number of folds was chosen to be 5 and 10, according to the common practice, providing objective accuracy assessment and reasonable computational requirements [44]. The original proportion of the classes in the complete dataset was also preserved within the created folds. Once the folds were created, the first fold was treated as a validation set, and the remaining K-1 folds were used for training. The class balancing and growing of the probability forest was performed only on the training set. The grown probability forest was then applied to obtain class probability predictions and later to perform spatio-temporal aggregation only on the validation set. This procedure was repeated K times; each time, a different fold was treated as a validation set. In order to suppress errors introduced by creating folds, the K-fold cross-validation procedure was performed 10 times, each time randomly creating new folds. Random folds were preferred over spatially independent folds because, for some classes, woody vegetation polygons had to be clustered, due to the specific environment and climate conditions required for these types of vegetation. Having in mind the structure of the input dataset, it would be difficult to create spatially independent folds. Therefore, although such a random split was not implicitly spatially independent, the authors believe that the same objective accuracy assessment was obtained due to the randomization and leave-location-out procedure. The evaluation was performed on the final merged dataset by creating the confusion matrix and calculating the overall accuracy (OA), producer's accuracy (PA), user's accuracy (UA) and kappa coefficient of agreement (Kappa). These metrics are considered good practices commonly used for evaluation of land cover classification results [42].

## 4. Results

The available woody vegetation polygons have been used to extract spectral information for each of the woody vegetation class. This resulted in 115,059 pixels representing all eight woody vegetation classes (Table 2). Since the classes in the created dataset are very unbalanced, the SMOTE method has been applied. The resulting dataset has almost perfectly balanced classes.

**Table 2.** The definition of classification scheme and the number of vectorized training polygons for their representation.

| Class ID | Informal Phytosociological Name | Extracted Number of Vegetation Pixels |
|:---:|:---:|:---:|
| 1 | Ostryo-Carpinion | 2299 |
| 2 | Quercion frainetto | 6705 |
| 3 | Quercion petraea-cerris | 11,152 |
| 4 | Quercion roboris | 65,561 |
| 5 | Fagion | 14,971 |
| 6 | Pinion nigrae | 3235 |
| 7 | Vaccinio-Piceion | 10,871 |
| 8 | Pinion mugo | 265 |

The probability forest has been grown using the created balanced dataset. The output model was then used to the predict class probability of each woody vegetation class for all Sentinel-2 forest pixels. This was done for both pixel-based and neighborhood-based approaches and for all three tested aggregation functions. The validation metrics show that the performance of the proposed woody vegetation classification method depends both on the aggregation function, as well as on the number of pixels used for the prediction (Table 3). All tested pixel-based aggregation functions achieved almost the same performance, where simple mean and geometric mean functions slightly outperformed the approach based on the most common class rule. The neighborhood-based performances were always higher than the performances of matching pixel-based methods. Increasing the size of the neighborhood had a limited effect, with only marginal improvements. Among the tested aggregation functions, the most common class rule provided the best performance of all tested combinations, but only in combination with the neighborhood-based approach. This suggests that this approach most successfully recognizes the situations of wrong predictions where the central pixel prediction is different from the predictions for the surrounding pixels.

**Table 3.** Validation metrics obtained after 5-fold and 10-fold leave-location out cross-validation.

| Aggregation Rule | | 5-Fold LLOCV | | 10-Fold LLOCV | |
|:---:|:---:|:---:|:---:|:---:|:---:|
| | | OA [%] | Kappa | OA [%] | Kappa |
| **pixel-based** | MC | 81.02 | 0.72 | 80.63 | 0.72 |
| | SM | 81.92 | 0.74 | 81.67 | 0.74 |
| | GM | 81.94 | 0.74 | 81.87 | 0.74 |
| $3 \times 3$ | MC ($3 \times 3$) | **83.09** | **0.76** | **82.89** | **0.75** |
| | SM ($3 \times 3$) | 82.27 | 0.74 | 81.68 | 0.74 |
| | GM ($3 \times 3$) | 82.39 | 0.74 | 81.87 | 0.74 |
| $5 \times 5$ | MC ($5 \times 5$) | **82.97** | **0.75** | **83.10** | **0.76** |
| | SM ($5 \times 5$) | 82.30 | 0.74 | 81.98 | 0.74 |
| | GM ($5 \times 5$) | 82.41 | 0.74 | 82.05 | 0.74 |

The supremacy of the most common class rule as an aggregation method is confirmed also by inspection of the Producer's and User's accuracy barplots (Figure 5). Tables with complete PA and UA values are available in the Appendix B.

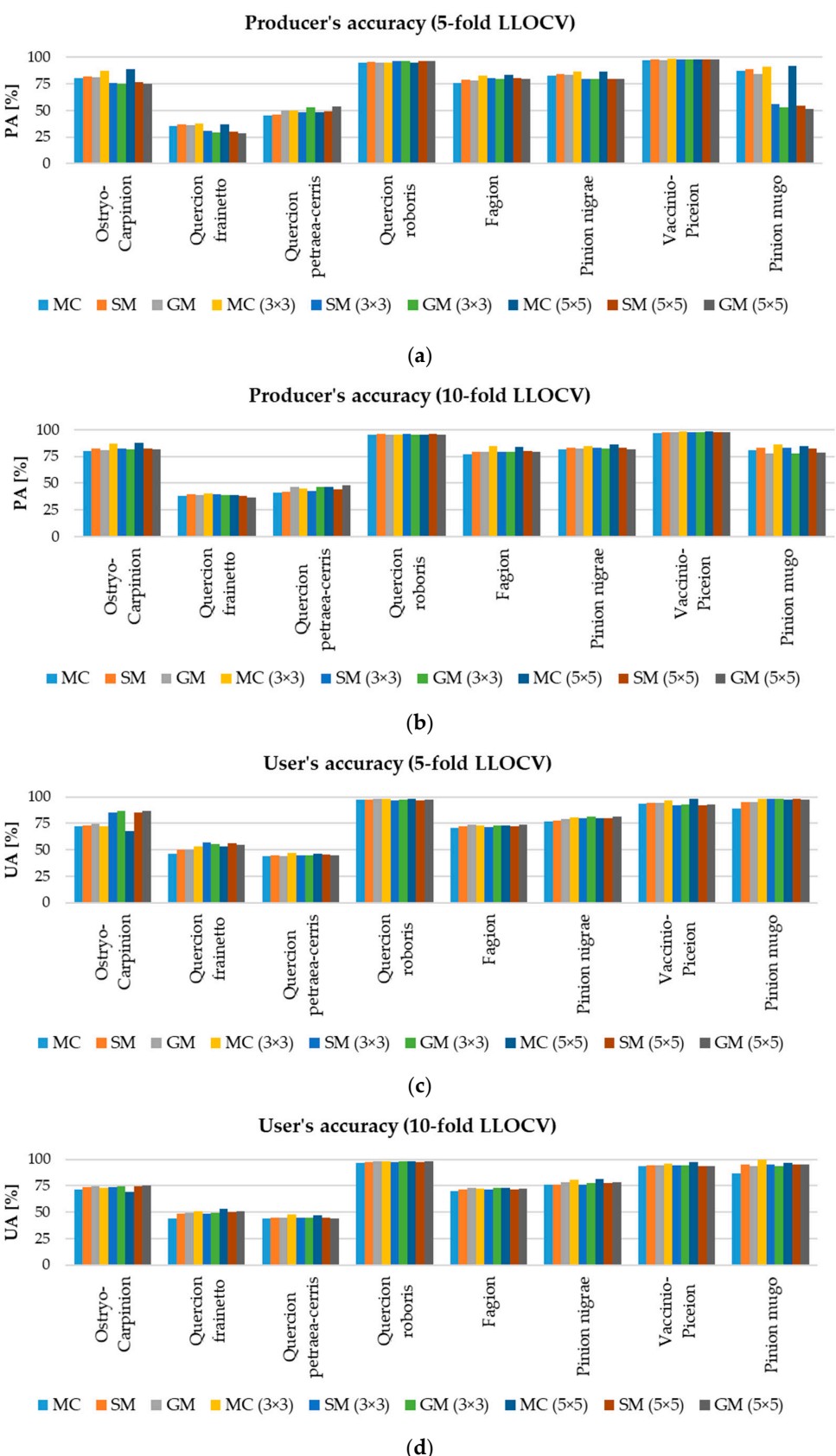

**Figure 5.** Producer's accuracy and user's accuracy barplots after K-fold leave-location out cross-validation (**a**) PA with K = 5 (**b**) PA with K = 10 (**c**) UA with K = 5 (**d**) UA with K = 10.

For most classes, the MC rule with either $3 \times 3$ or $5 \times 5$ neighborhood provides the best metrics or close to the best metrics. For half of the modeled types of woody vegetation, the difference in the performance of this approach is much greater than the rest of the methods. This is evident for coniferous woody vegetation (Pinion nigrae, Vaccinio-Piceion and Pinion mugo) with PA being higher from 2% up to 40% and with the preservation of high UA (greater than 80%). Improvement is also present for Fagion woody vegetation, although it is not as large as for coniferous woody vegetation. These four woody vegetation classes represent more than a third of the total forest area, which suggests that such improvements have a major impact on the final classification map. It is important to note that these improvements are obtained even with a considerable disproportion of classes in the original training dataset. This highlights the importance of class balancing using SMOTE, as some of these classes are minority classes (Pinion mugo) and some have a very large number of training observations (Fagion). This suggests that the improvements of this approach are mainly due to the ability to capture the specifics of these types of woody vegetation growing in large, homogenous and dense groups, without "small islands" of different types of woody vegetation. It should be noted that for some other woody vegetation classes, the neighborhood-based MC rule does not perform the best (like for Ostryo-Carpinion), where it provides the highest PA, but fails to provide the best UA.

The differences between five-fold and 10-fold LLOCV validation metrics are least noticeable for the MC rule in combination with the neighborhood approach. In contrast, performances of all other methods change with different numbers of folds. This is particularly evident for the Pinion mugo class. The MC rule with both neighborhoods provides the best metrics for its prediction, indicating significantly higher accuracy than other neighborhood-based aggregations in case of five-fold LLOCV. In case of 10-fold LLOCV, it is still the best approach, but without large differences in performances. This can be explained by the amount of data used for model training. In every step, 10-fold LLOCV uses ≈90% of the data of each class for model training, compared to ≈80% used by five-fold LLOCV. Even though the SMOTE is applied, this shows that such reduction can severely influence the model performances, particularly for minority classes (as Pinion mugo). This shows that the MC rule in combination with the neighborhood approach is the most robust regarding the amount of training data necessary for proper modeling of minority classes.

All validation metrics indicate that the best results are achieved by using the MC rule, with almost the same performance of both neighborhoods. Individually per woody vegetation class, the $5 \times 5$ neighborhood outperforms the $3 \times 3$ neighborhood for most of the classes for both PA and UA. Confusion matrix after five-fold and 10-fold LLOCV are given in Appendix C. Therefore, this approach can be considered to be the best one for woody vegetation classification using the proposed spatio-temporal framework. Consequently, the final classification map has been produced using the $5 \times 5$ neighborhood with the MC rule (Figure 6). The covered areas of each woody class are given in the Table 4.

**Table 4.** Area and share in the total forest area for each woody vegetation class after classification using the MC rule and $5 \times 5$ neighborhood for aggregation.

| Class ID | Informal Phytosociological Name | Area [km²] | Area [%] |
|:---:|:---:|:---:|:---:|
| 1 | Ostryo-Carpinion | 4411 | 12.42 |
| 2 | Quercion frainetto | 1104 | 3.11 |
| 3 | Quercion petraea-cerris | 11,707 | 32.95 |
| 4 | Quercion roboris | 5528 | 15.56 |
| 5 | Fagion | 10,768 | 30.31 |
| 6 | Pinion nigrae | 949 | 2.67 |
| 7 | Vaccinio-Piceion | 907 | 2.55 |
| 8 | Pinion mugo | 153 | 0.43 |

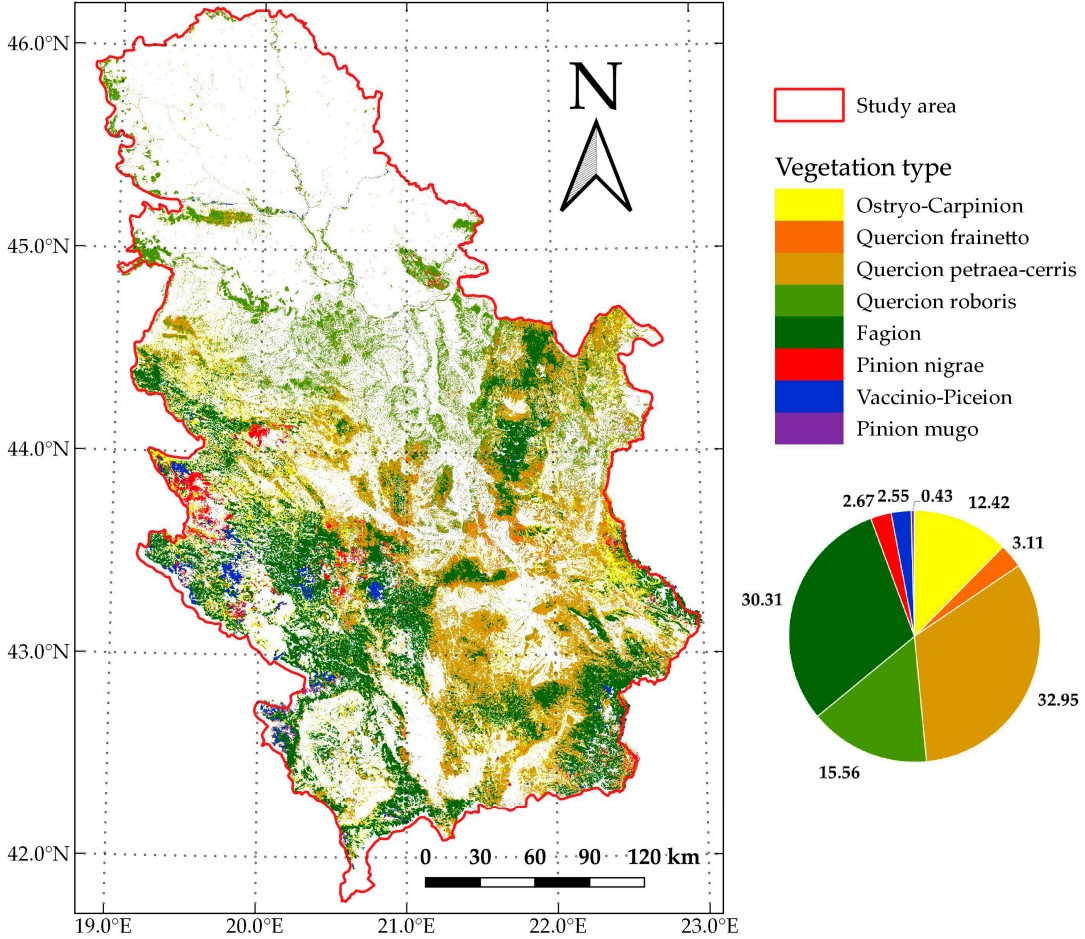

**Figure 6.** Final classification map of woody vegetation coverage over the study area.

## 5. Discussion

### 5.1. Final Classification Map of Woody Vegetation

The previously explained LLOCV procedure can give information about the model accuracy, but the accuracy metrics are generally biased for mapping the entire study area. Unfortunately, no comparable external data source is available for the purpose of assessing the accuracy of the final woody vegetation map. In addition, creating a validation set based on probability sampling requires extensive resources and field work, which are beyond the scope of this research. Therefore, validation of the final classification map of woody vegetation has been performed by using visual inspection and comparison with available statistical reports.

By visual inspection and comparison, the final classification map of woody vegetation produced in this paper corresponds to the existing general vegetation maps: Map of the Natural Vegetation of Europe (scale 1:2,500,000) [29] and the Map of Natural Potential Vegetation of SFR Yugoslavia (scale 1:1,000,000) [45]. Additionally, the proportional relationships of the analyzed vegetation types for most of the classes fit into the results obtained by the first National Forest Inventory of Serbia in the period 2004–2006, which was done using traditional field survey methods that provide a highly accurate forest inventory [46,47]. Significant deviations were obtained only in the proportions of thermophytic broadleaved deciduous forests and mesophytic and hygromesophytic broadleaved forests. The share of Ostryo-Carpinion forests in this research is 12.42%, compared to the share of *Carpinus orientalis* and *Ostrya carpinifolia* forests in [47], which is 3.9%. The results of this research show that Quercion roboris forests cover 16% of the area, while the corresponding forest stand category in [47] *Quercus robur* forests has the coverage of 1.4%.

In the first case (Ostryo-Carpinion), the discrepancy can be explained by the fact that the data from the National Forest Inventory of Serbia refer primarily to forests and do not include data on shrubs and lower degradation forms of forest vegetation. Since in the thermophilic forest zone, the degradation forms of forest vegetation are mainly represented by shrub vegetation belonging to the Ostryo-Carpinion type, the increased share of this type of vegetation in our research can be explained by the assumption that different degradation forms of thermophytic broadleaved were recognized in this class. In the second case (Quercion roboris), the deviation is explained by the fact that our class Quercion roboris is much wider than the class *Quercus robur* forests presented in the first National Forest Inventory of Serbia (Appendix A).

Since the spatial distribution of the training polygons cannot be considered ideal, it is possible that some discrepancies in the area assessments are also caused as a result. Therefore, additional woody vegetation polygons in the forest areas sparsely covered by polygons (like southern part of the study area) can significantly benefit in better modeling classes in RF model and more accurate area assessment.

*5.2. Variable Usefulness and Importance*

The usefulness and importance of variables used for the woody vegetation classification is determined first by inspection of the temporal phenological spectral patterns. Temporal phenological spectral patterns are created by plotting the average spectral responses of all woody vegetation types per each available date (Figure 7). This was done for all used Sentinel-2 spectral bands and the calculated NDVI value.

The created temporal phenological spectral plots show that patterns vary per woody vegetation type and per spectral band. The decreasing pattern can be observed across most of the bands. This is as expected, since all woody vegetation classes are in full vegetation at the start of the observation period, which later decreases as autumn and winter approach. This decreasing pattern is particularly noticeable for deciduous tree species, while it is less present for coniferous tree species. The separability between species also follows such a pattern, with the highest separability being in the June–August period. This separability highly depends on the type of woody vegetation. Independently per band, Bands B11 (Short-wave infrared 1) and B12 (Short-wave infrared 2) visually show the highest separability between types of woody vegetation. Deciduous and coniferous woody vegetation groups can be clearly observed across these spectral bands. All three red edge bands (B05, B06 and B07), as well as the NIR band, also show high spectral separability between vegetation types. Individual vegetation types also show specific distinguishing patterns on certain bands, such as Fagion woody vegetation on bands B02 (Blue) and B04 (Red) or Ostryo-Carpinion on band B03 (Green). These can be potentially used to subset spectral bands and further relieve the model complexity. Ottosen et al. also highlighted that the classification results are improved by using only subsets of Sentinel-2 spectral bands [48]. This was not taken into account in this research, even though it is important to understand how the classification results obtained by this methodology are influenced by the subset of used spectral bands. This needs to be addressed in the future.

The variable importance is additionally assessed through the Mean decrease Gini measure (Figure 8). The Mean decrease Gini is a measure of how much the Gini impurity metric is reduced by a variable for each class. The Mean decrease Gini metric was chosen as the importance measure because it was also used for growing the probability random forest. The scaled variable importance based on the Mean decrease Gini measure is produced and plotted. The results correspond to the observed temporal phenological spectral patterns. Bands B11 and B02 are the most important spectral variables, followed by B12. The day and month temporal variables turned out to be very important, being the second and third most important variables of all. This highlights the model's ability to capture woody vegetation phenological patterns. The other spectral variables are of approximately the same importance.

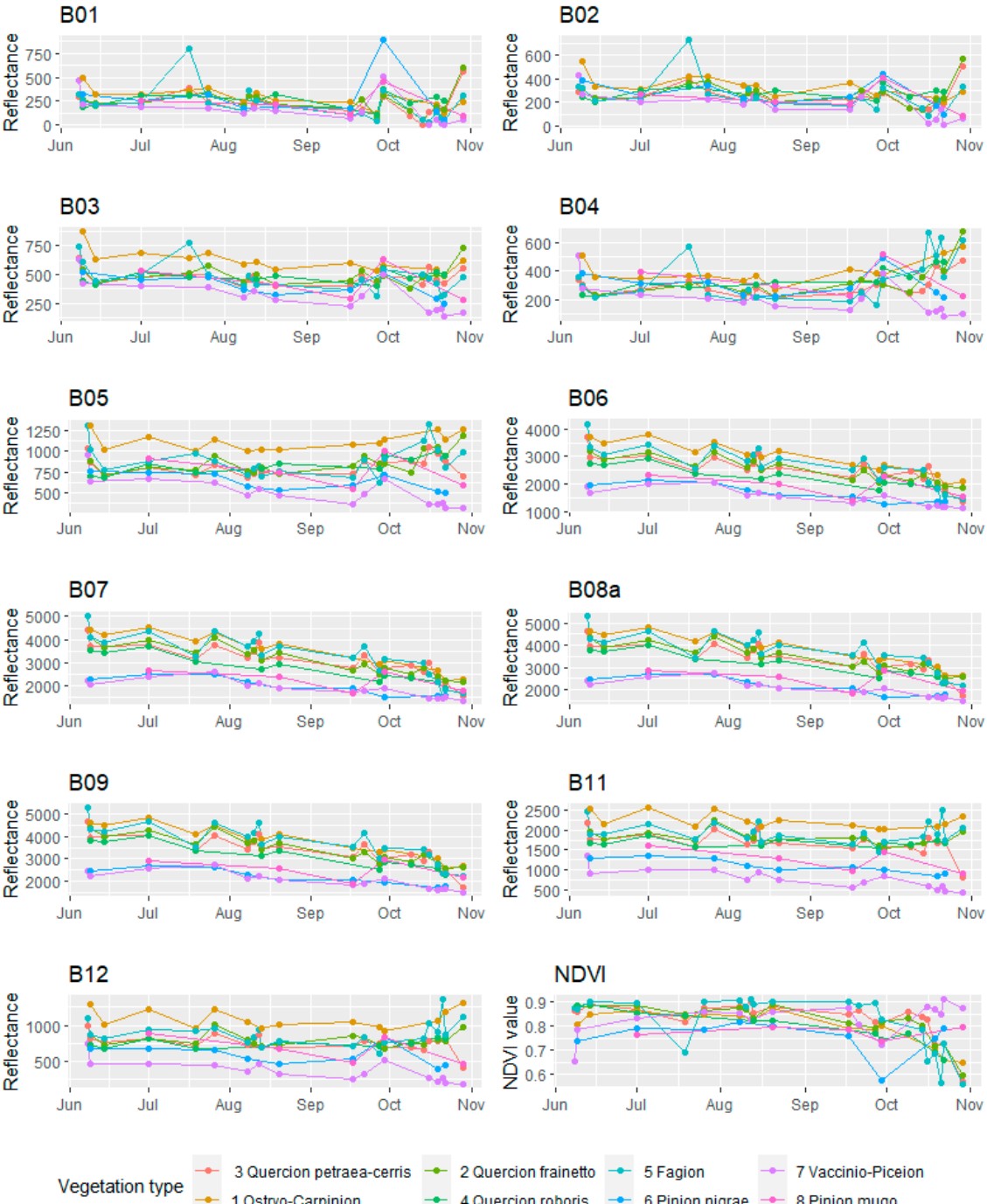

**Figure 7.** Temporal phenological patterns for the used Sentinel-2 spectral bands and NDVI values.

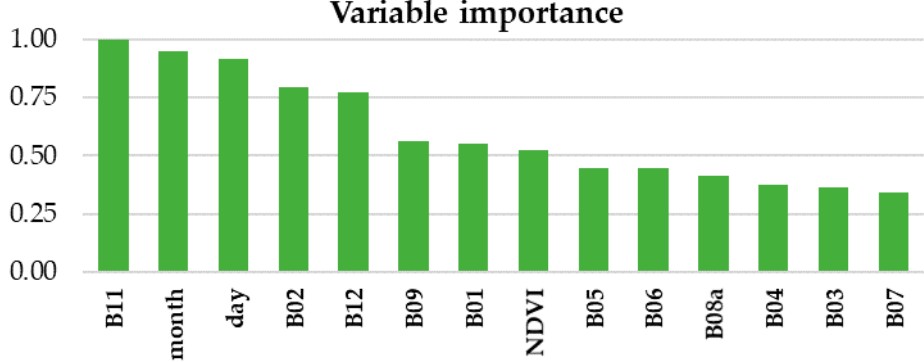

**Figure 8.** Variable importance barplot expressed through the Mean decrease Gini measure.

*5.3. Spatio-Temporal Framework for Woody Vegetation Classification*

The spatio-temporal framework proposed in this research defines the timestamp of the observation explicitly in the model through day and month variables. Such change in the model means that, in order to determine the woody vegetation class, the methodology requires only one cloud-free pixel available for each part of the study area. This change in the model allows timestamps to be freely combined and enables the entire methodology to be easily adjusted to any area of interest, regardless of its size. Additionally, the model is lighter, as every spectral band is represented only once through a single variable, as an alternative to multiple timestamp–spectral band variable combinations. There is also no data-wrangling and only the original observations are used for both training and prediction phases. As the number of cloud-free pixels over the study area increases, there will be multiple predictions for each pixel of the study area. This enables the full potential of spatio-temporal aggregation to be exploited. The spatio-temporal aggregation is not only pixel-based, but is expanded to take into account the neighboring pixel predictions as well. The main idea behind the neighborhood-based approach is to solve situations where surrounding pixels show strong probability of belonging to a certain class, which is different from the class probability of the central pixel. In these cases, there is a strong reason to believe that the central pixel should also belong to the class of surrounding pixels. It is more likely that the prediction based solely on spectral information for the central pixel is wrong, which is a consequence of some noise present in the spectral information for that pixel, than the probability that the central pixel's class differs from the class of its neighbors.

Applied spatio-temporal aggregation can be affected by temporal and/or spatial homogeneity. For areas with no temporal and spatial heterogeneity over the time period covered with satellite observations and the used neighborhoods, the prediction results should be very reliable. In other cases, the proposed methodology can deal with such heterogeneity up to a certain amount. This is because each of the used S2 observation influences the decision on the final woody vegetation class and any temporal or spatial change can potentially suppress other observations. Temporally, this means that changes that last longer within the used time interval are more likely to be recognized by the model. Similarly, only spatial changes that have larger spatial extents compared to the neighborhood size will be recognized. This means that the minimum mapping unit gets coarser as the size of neighborhood increases. At the same time, the neighborhood approach achieves smoothing effects, where isolated pixels are classified into the class of pixels in the neighborhood. Therefore, it is not advised to extend both the observation period and the neighborhood size more than necessary. A shorter observation period reduces the amount of changes that occur in the first place, and the appropriate neighborhood size limits the amount of smoothing which occurs.

The proposed methodology is applied using resampled 60 m Sentinel-2 bands. This spatial resolution may be too coarse, since the increase in neighborhood size has a limited effect on the performance. It is expected that using the original spatial resolution of 10 m and 20 m for available bands, should reduce the effects of mixed pixels and further increase the classification accuracy. In addition, this should help the previously discussed heterogeneity issues, where smaller changes

could be modelled. Stronger neighborhood effects can also be expected to occur for higher spatial resolutions, so the neighborhood size could become a more important parameter in the proposed methodology. However, there are several motivations for choosing a coarser spatial resolution of 60 m for all bands. Firstly, this is done in order to match the chosen classification scheme, designed for country-wide woody vegetation mapping. The used classification scheme is defined according to the "Database on the distribution of potentially endangered species and habitats of Serbia" and the increase in spatial resolution would require the introduction of new woody vegetation classes for which the proper training data are lacking. Secondly and more importantly, the proposed methodology aims to be used on large areas and this way the computation and performance issues are reduced. At a spatial resolution of 60 m, the used dataset comprises more than 160 million S2 observations, which would increase about nine or 36 times in the case of 20 m and 10 m spatial resolutions, respectively. One solution to deal with such an increase in size of data is to select only more informative spectral bands. Since the proposed methodology has advantages and applicability, especially over larger areas, this can become a very important issue.

The results of this research match those obtained by other similar studies, which were based on high- or medium-resolution optical data. Grabska et al. have achieved an OA greater than 90% on the study area of 305 km$^2$ for nine tree species using 18 cloud-free sentinel-2 images [6]. Liu et al. identified forest types using Sentinel-1A, Sentinel-2A, Landsat-8 and DEM Data with OA of 82.78%. They extracted the four single-dominant forests and the four mixed forest types over the area of 2261 km$^2$ [40]. Lu et al. implemented Spatial–Spectral–Temporal Integrated Fusion to identify six tree species and two mixed forest types with an OA of 83.6% over the area of 1610 km$^2$ [49]. In addition to covering a significantly larger study area, this research and the proposed solution also completely relies only on optical satellite sources. Therefore, there is a potential for further improvement by using other data sources such as radar satellite data, LiDAR, multiple optical sources and/or other auxiliary data (topography, etc.).

This proposed spatio-temporal framework relies on Sentinel-2 scene classification maps and a forest mask to properly identify forest areas. This is potentially the weak point of the method, because the output quality is in a way limited by the quality of the used forest mask. It is important to use the most recent available forest mask as possible, especially in the areas of rapid forest change. The forest mask used in this research is from 2015, which creates a 4-year gap with the used Sentinel-2 data. Such a forest mask can be considered outdated since there is an active forest management over the study area, with the forest area being influenced by felling of trees, forest growing and silviculture. Additionally, some forest fires occur annually, usually in the warm summer period. However, these changes are not extensive and do not have large spatial extents. Visual inspection of forest mask showed that the misrepresentations happen only in a few local areas, so the authors believe that these errors do not significantly affect the overall results. The methodology can also be further expanded to address this issue, with the same spatio-temporal framework being used to firstly identify the forest and non-forest pixels. This issue was beyond the scope of this research due to the lack of proper training data for good representation of non-forest areas. This issue needs to be addressed in the future. Another setback of the proposed approach is that each pixel is always assigned to some class. In a situation where all woody vegetation types are well represented by training polygons, this does not pose a problem; however, when the study area is very complex or insufficiently researched in terms of woody vegetation, it might not be possible to represent all the woody vegetation types in the study area.

The study has revealed several problematic cases in which predictions do not achieve a satisfactory level of accuracy and reliability. Some of them are problems with: (1) identification of sufficiently representative learning polygons within the complex and mosaic area, such as alluvial forests in riparian and swamp galleries and woodland in alluvial plains; (2) insufficiently researched natural vegetation types, such as forests of birch (Betula pendula), common aspen (Populus tremula) or rowan (Sorbus aucuparia); (3) the natural absence of clear boundaries between ecologically and physiognomically close vegetation types that build phylogenetically close tree species (e.g., gradual

transition between thermophytic (Quercion frainetto), thermo-mesophytic (Quercion petraea-cerris) and mesophytic to hygromesophytic (Quercion roboris) forests in which all three main oak species have such a wide ecological valence that their populations occur within all three types of oak forests); (4) a priori classification of mixed deciduous and coniferous woodlands in which a large number of tree species have different parts in the general coverage; (5) an absence of data on areas with highly artificial tree plantations or naturalized forest of alien species(e.g., black locust forests (Robinia pseudoaccacia), american ash (*Fraxinus* spp.) and maple (*Acer* spp.) forests, poplar plantations (*Populus* spp.), etc.

Given that eight analyzed woody vegetation classes cover a large portion of the study area, it can be expected that the omission of other classes should not significantly affect the spatial coverage of woody vegetation in the study area. Since there are multiple predictions for each pixel of the study area, as well as the probability of each prediction, there is a possibility to use this information to identify the weak pixels, which probably do not belong to any of the classes. This is also something to be considered in future research.

## 6. Conclusions

The novel spatio-temporal classification framework has been presented in this paper. The proposed methodology has been successfully applied for mapping eight potential types of woody vegetation over the study area of Serbia. This was done using 88 Sentinel-2 multitemporal granules dating from the June–October period of 2019. The method relies on forest mask to exclude non-forest areas and on training polygons to represent each woody vegetation class of interest. The used framework is novel because it includes temporal information explicitly in the probability random forest model that has been grown, which produces several probability predictions for each pixel of the study area. Multiple predictions enable specific spatio-temporal aggregation to be performed in order to determine most probable woody vegetation class for each pixel of the study area.

Several spatio-temporal aggregation approaches have been tested within this research. They can be divided into pixel-based and neighborhood-based methods. According to the validation metrics, the most common class rule together with the neighborhood of $5 \times 5$ pixels provided the best results. The most common class rule means that from the set of multiple predictions, each prediction casts a vote to the class with the highest probability. The final class is then set to the class that has the highest number of votes. The overall accuracy and kappa coefficient of agreement obtained from five-fold cross validation by this approach are 82.97% and 0.75, respectively. The corresponding producer's accuracies range from 36.74% to 97.99% and user's accuracies range from 46.31% to 98.43%. Such results match the performances of other similar studies, although most of them have been done for much smaller study areas.

Given the obtained accuracy, it can be said that the proposed methodology is applicable for woody vegetation mapping in Serbia. The classification scheme and the corresponding training polygons are prerequisites for its implementation. The methodology has been tested over the territory of Serbia, but it has the potential to be transferred to other areas if appropriate adjustments to the classification scheme are introduced and new training polygons are made available. Further testing is necessary to confirm such assumptions. It is implemented in such a way that it can be easily integrated into existing operational programs intended for forest inventory and mapping. The identified weak points of the framework are its reliability on the forest mask and the fact that each pixel of the study area is always assigned to some class. The second issue is more complex and maybe it can be overcome by using the probability of each prediction to identify the weak probability pixels, indicating that they probably do not belong to any of the classes. These two things are planned to be addressed in future research.

**Author Contributions:** All authors worked equally on the research conceptualization, investigation, discussion, and data modeling. J.K. was involved in data preparation and case study implementation. D.L., N.K. and J.Š.-S. covered the biological aspects of the research. All authors participated in the writing of the manuscript, however J.K. took the lead. All authors have read and agreed to the published version of the manuscript.

**Funding:** This research received no external funding.

**Acknowledgments:** This study was supported by the Serbian Ministry of Education, Science and Technological Development, projects TR 36020 and OI 173030. The authors also thank the anonymous reviewers for constructive and valuable comments that improved the quality of this study.

**Conflicts of Interest:** The authors declare no conflict of interest.

## Appendix A

**Table A1.** Analyzed types of forests and shrubs according to the national classification of habitats in Serbia.

| Class ID | SrbHab Code | Habitat Type Name |
|---|---|---|
| 1 | A2.9 | Forests of Oriental hornbeam (*Carpinus orientalis*) and hop-hornbeam (*Ostrya carpinifolia*) |
| | A2.91 | Forests of Oriental hornbeam (*Carpinus orientalis*) |
| | A2.92 | Forests of hop-hornbeam (*Ostrya carpinifolia*) |
| 2 | A2.1 | Forests of Hungarian (*Quercus frainetto*) and Austrian oak (*Quercus cerris*) forests |
| | A2.11 | Forests of Hungarian (*Quercus frainetto*) and Austrian oak (*Quercus cerris*) forests |
| | A2.12 | Forests of Hungarian oak (*Quercus frainetto*) |
| 3 | A2.5 | Forests of sessile (*Quercus petraea*) and Austrian oak (*Quercus cerris*) |
| | A2.51 | Forests of sessile oak (*Quercus petraea*) |
| | A2.52 | Forests of Austrian oak (*Quercus cerris*) |
| 4 | A1.3 | Forests of pedunculate oak (*Quercus robur*) and field ash (*Fraxinus angustifolia*) |
| | A1.31 | Forests of pedunculate oak (*Quercus robur*) |
| | A1.33 | Forests of pedunculate oak (*Quercus robur*) and field ash (*Fraxinus angustifolia*) |
| | A1.35 | Forests of pedunculate oak (*Quercus robur*), hornbeam (*Carpinus betulus*) and field ash (*Fraxinus angustifolia*) |
| | A1.4 | Forests of pedunculate oak (*Quercus robur*) and hornbeam (*Carpinus betulus*) |
| | A1.41 | Forests of pedunculate oak (*Quercus robur*) and hornbeam (*Carpinus betulus*) |
| | A1.42 | Forests of pedunculate oak (*Quercus robur*), hornbeam (*Carpinus betulus*) and Austrian oak (*Quercus cerris*) |
| 5 | A3.2 | Beech forests (*Fagus moesiaca*) |
| | A3.22 | Submontane beech forests (*Fagus moesiaca*) |
| | A3.23 | Mountain beech forests (*Fagus moesiaca*) |
| | A3.27 | Subalpine beech forests (*Fagus moesiaca*) |
| 6 | A5.1 | Forests of black (*Pinus nigra*) and white pine (*Pinus sylvestris*) |
| | A5.11 | Black pine forest (*Pinus nigra*) |
| | A5.12 | Forest of black (*Pinus nigra*) and white pine (*Pinus silvestris*) |
| 7 | A6.1 | Spruce (*Picea* spp.) and fir (*Abies* spp.) Forests |
| | A6.11 | Spruce (*Picea abies*) and fir (*Abies alba*) forests |
| | A6.12 | Spruce forests (*Picea abies*) |
| 8 | B6.1 | Krummholz (*Pinus mugo*) scrub |

## Appendix B

**Table A2.** Producer's accuracy for all woody vegetation classes for 5-fold LLOCV.

| Vegetation Type | MC | SM | GM | MC (3 × 3) | SM (3 × 3) | GM (3 × 3) | MC (5 × 5) | SM (5 × 5) | GM (5 × 5) |
|---|---|---|---|---|---|---|---|---|---|
| Ostryo-Carpinion | 80.12 | 81.93 | 80.84 | 86.88 | 76.18 | 75.26 | 88.48 | 76.22 | 75.24 |
| Quercion frainetto | 35.59 | 37.17 | 36.31 | 37.73 | 30.41 | 29.29 | 36.74 | 29.87 | 28.73 |
| Quercion petraea-cerris | 45.25 | 46.34 | 50.04 | 49.70 | 48.65 | 53.00 | 48.54 | 49.13 | 53.44 |
| Quercion roboris | 95.17 | 95.32 | 94.94 | 94.92 | 96.55 | 96.30 | 94.88 | 96.59 | 96.33 |
| Fagion | 76.15 | 78.88 | 78.12 | 82.82 | 80.54 | 79.45 | 83.61 | 80.51 | 79.37 |
| Pinion nigrae | 82.46 | 84.06 | 83.29 | 86.58 | 79.82 | 79.40 | 86.12 | 79.85 | 79.43 |
| Vaccinio-Piceion | 97.18 | 97.57 | 97.48 | 98.53 | 98.21 | 98.02 | 97.99 | 98.20 | 98.00 |
| Pinion mugo | 87.16 | 88.69 | 84.25 | 91.44 | 55.96 | 52.75 | 91.98 | 54.17 | 51.54 |

**Table A3.** Producer's accuracy for all woody vegetation classes for 10-fold LLOCV.

| Vegetation Type | MC | SM | GM | MC (3 × 3) | SM (3 × 3) | GM (3 × 3) | MC (5 × 5) | SM (5 × 5) | GM (5 × 5) |
|---|---|---|---|---|---|---|---|---|---|
| Ostryo-Carpinion | 80.23 | 82.16 | 81.11 | 86.88 | 82.16 | 81.15 | 87.35 | 82.24 | 81.44 |
| Quercion frainetto | 37.81 | 39.29 | 38.57 | 40.60 | 39.30 | 38.58 | 38.42 | 37.80 | 36.83 |
| Quercion petraea-cerris | 40.95 | 42.22 | 46.32 | 44.52 | 42.45 | 46.49 | 46.61 | 44.23 | 48.01 |
| Quercion roboris | 95.59 | 95.97 | 95.60 | 95.43 | 95.95 | 95.58 | 95.25 | 95.75 | 95.35 |
| Fagion | 76.76 | 79.48 | 79.33 | 84.51 | 79.45 | 79.29 | 84.15 | 79.67 | 79.23 |
| Pinion nigrae | 81.17 | 82.93 | 82.10 | 84.67 | 82.93 | 82.10 | 85.81 | 82.79 | 81.82 |
| Vaccinio-Piceion | 97.13 | 97.42 | 97.52 | 98.63 | 97.42 | 97.51 | 98.48 | 97.68 | 97.58 |
| Pinion mugo | 80.65 | 83.04 | 77.61 | 85.87 | 83.04 | 77.61 | 84.83 | 82.07 | 78.39 |

**Table A4.** User's accuracy for all woody vegetation classes for 5-fold LLOCV.

| Vegetation Type | MC | SM | GM | MC (3 × 3) | SM (3 × 3) | GM (3 × 3) | MC (5 × 5) | SM (5 × 5) | GM (5 × 5) |
|---|---|---|---|---|---|---|---|---|---|
| Ostryo-Carpinion | 71.90 | 73.29 | 74.12 | 72.56 | 85.39 | 86.50 | 67.54 | 85.39 | 86.44 |
| Quercion frainetto | 46.21 | 50.35 | 50.06 | 53.11 | 56.84 | 55.60 | 53.17 | 56.09 | 54.84 |
| Quercion petraea-cerris | 43.79 | 44.55 | 43.83 | 47.18 | 44.94 | 44.45 | 46.31 | 45.17 | 44.58 |
| Quercion roboris | 97.02 | 97.27 | 97.95 | 98.29 | 96.30 | 97.23 | 98.43 | 96.27 | 97.20 |
| Fagion | 70.46 | 72.25 | 73.62 | 73.09 | 71.79 | 73.18 | 73.32 | 72.05 | 73.46 |
| Pinion nigrae | 76.96 | 77.87 | 78.95 | 80.90 | 79.84 | 81.01 | 80.07 | 80.02 | 81.13 |
| Vaccinio-Piceion | 93.57 | 94.25 | 94.33 | 96.27 | 92.25 | 92.48 | 97.86 | 92.30 | 92.53 |
| Pinion mugo | 89.20 | 95.39 | 95.00 | 98.36 | 98.12 | 98.01 | 97.39 | 98.04 | 97.66 |

**Table A5.** User's accuracy for all woody vegetation classes for 10-fold LLOCV.

| Vegetation Type | MC | SM | GM | MC (3 × 3) | SM (3 × 3) | GM (3 × 3) | MC (5 × 5) | SM (5 × 5) | GM (5 × 5) |
|---|---|---|---|---|---|---|---|---|---|
| Ostryo-Carpinion | 71.56 | 73.51 | 74.71 | 72.72 | 73.47 | 74.65 | 69.19 | 74.18 | 74.96 |
| Quercion frainetto | 43.75 | 48.88 | 49.45 | 50.63 | 48.85 | 49.40 | 52.89 | 50.20 | 50.43 |
| Quercion petraea-cerris | 43.67 | 44.69 | 44.33 | 47.61 | 44.82 | 44.43 | 46.80 | 44.98 | 44.26 |
| Quercion roboris | 96.97 | 97.13 | 97.90 | 98.22 | 97.18 | 97.94 | 98.30 | 97.30 | 97.98 |
| Fagion | 69.86 | 71.51 | 72.64 | 72.39 | 71.47 | 72.58 | 72.76 | 71.31 | 72.47 |
| Pinion nigrae | 75.93 | 76.14 | 77.92 | 80.43 | 76.14 | 77.89 | 81.67 | 77.59 | 78.59 |
| Vaccinio-Piceion | 93.36 | 94.13 | 94.20 | 95.70 | 94.12 | 94.20 | 97.07 | 93.57 | 93.84 |
| Pinion mugo | 86.89 | 94.79 | 93.46 | 99.50 | 94.79 | 93.46 | 96.60 | 95.45 | 95.25 |

## Appendix C

**Table A6.** Confusion matrix after the 5-fold LLOCV using the MC rule and 5 × 5 neighborhood for aggregation (Legend for Class ID: 1 Ostryo-Carpinion, 2 Quercion frainetto, 3 Quercion petraea-cerris, 4 Quercion roboris, 5 Fagion, 6 Pinion nigrae, 7 Vaccinio-Piceion, 8 Pinion mugo).

| | | Reference | | | | | | | | |
|---|---|---|---|---|---|---|---|---|---|---|
| | Class ID | 1 | 2 | 3 | 4 | 5 | 6 | 7 | 8 | Total |
| Prediction | 1 | 4309 | 190 | 858 | 15 | 288 | 676 | 27 | 17 | 6380 |
| | 2 | 9 | 4495 | 2989 | 339 | 621 | 0 | 0 | 1 | 8454 |
| | 3 | 215 | 4589 | 11,442 | 4442 | 3765 | 206 | 48 | 3 | 24,710 |
| | 4 | 1 | 621 | 796 | 88,903 | 0 | 1 | 0 | 0 | 90,322 |
| | 5 | 336 | 2121 | 6247 | 0 | 23,979 | 0 | 22 | 0 | 32,705 |
| | 6 | 0 | 218 | 1024 | 0 | 5 | 5710 | 154 | 20 | 7131 |
| | 7 | 0 | 0 | 205 | 0 | 20 | 34 | 12,347 | 11 | 12,617 |
| | 8 | 0 | 0 | 11 | 0 | 0 | 3 | 2 | 596 | 612 |
| | Total | 4870 | 12,234 | 23,572 | 93,699 | 28,678 | 6630 | 12,600 | 648 | 182,931 |
| | PA [%] | 88.48 | 36.74 | 48.54 | 94.88 | 83.61 | 86.12 | 97.99 | 91.98 | |
| | UA [%] | 67.54 | 53.17 | 46.31 | 98.43 | 73.32 | 80.07 | 97.86 | 97.39 | |

**Table A7.** Confusion matrix after the 10-fold LLOCV using MC rule and 5 × 5 neighborhood for aggregation (Legend for Class ID: 1 Ostryo-Carpinion, 2 Quercion frainetto, 3 Quercion petraea-cerris, 4 Quercion roboris, 5 Fagion, 6 Pinion nigrae, 7 Vaccinio-Piceion, 8 Pinion mugo).

| | | Reference | | | | | | | | |
|---|---|---|---|---|---|---|---|---|---|---|
| | Class ID | 1 | 2 | 3 | 4 | 5 | 6 | 7 | 8 | Total |
| Prediction | 1 | 4254 | 157 | 702 | 8 | 343 | 637 | 27 | 20 | 6148 |
| | 2 | 16 | 4606 | 2980 | 353 | 753 | 0 | 0 | 1 | 8709 |
| | 3 | 260 | 4123 | 10,485 | 3921 | 3379 | 191 | 41 | 5 | 22,405 |
| | 4 | 0 | 681 | 801 | 85,803 | 2 | 2 | 0 | 0 | 87,289 |
| | 5 | 340 | 2207 | 6394 | 1 | 23,916 | 0 | 10 | 1 | 32,869 |
| | 6 | 0 | 216 | 826 | 0 | 7 | 5257 | 113 | 18 | 6437 |
| | 7 | 0 | 0 | 295 | 0 | 22 | 37 | 12,408 | 21 | 12,783 |
| | 8 | 0 | 0 | 10 | 0 | 0 | 2 | 1 | 369 | 382 |
| | Total | 4870 | 11,990 | 22,493 | 90,086 | 28,422 | 6126 | 12,600 | 435 | 177,022 |
| | PA [%] | 87.35 | 38.42 | 46.61 | 95.25 | 84.15 | 85.81 | 98.48 | 84.83 | |
| | UA [%] | 69.19 | 52.89 | 46.80 | 98.30 | 72.76 | 81.67 | 97.07 | 96.60 | |

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
