# Peer review of "Spatio-Temporal Classification Framework for Mapping Woody Vegetation from Multi-Temporal Sentinel-2 Imagery"

_remotesensing, doi:10.3390/rs12172845_

Round 1

Reviewer 1 Report

In this study, the authors present a new spatio-temporal classification framework for mapping woody vegetation from Sentinel-2 multitemporal images. The method defines the variables in the model and uses the neighborhood of pixels to determine the most likely class of woody vegetation. However, it presents a confusing methodology when developing the process and the results are difficult to understand.

Observations generals

  1. Adapt the format of the journal in the sections "Materials and methods"
  2. It is recommended to add north, scale and coordinate system in Figures 1,2b, 6.
  3. The methodology presents deficiencies in the process, which can confuse the reader, for example, the kappa coefficient is spoken as a metric to evaluate the classification in the results and this should be explained in the methodology.

Introduccion

Line 49. What are the causes of these challenges, based on what, this claim is made.

Line 68Mention what those other ways are. Support with references

Materials

Line 100. It is recommended to mention the type of forests

Line 126. What date is the basemap?, since the phenology of plant species affects the selection of polygons

  1. From 2015 to 2019, did the vegetation polygons not change?, were there forest fires, forest harvesting or forest management?

Methodology

Line 178. What is the NDVI?

Line 193. Why not use band 8

  1. Because band 8 is excluded
  2. What percentage of the data were for training and testing in the cross-validation? 10 is the number of interactions, explain better.

Reviewer 2 Report

The manuscript applies the probability random forest (PRF) method on an annual collection of 60-m Sentinel-2 (L2A - BOA) images to classify country-wide forest types. The timestamp info is included directly in the model that is trained and validates using manually digitized polygons form high-resolution satellite images. Three different ways of aggregating the individual PRF predictions are then analyzed to conclude about the final prediction per pixel. The PRF model and the selected aggregation strategy are applied to derive a country-wide map of forest types. The classification accuracy is also assed and reported.

The topic fits to the scope of the journal, and the metrology applied is generally sound. However, certain aspects of the methodological setting and the validation approach should be explained in more detail and discussed directly in the manuscript. Therefore, I would recommend the manuscript for publication, but before that the comments raised here should be addressed. The general and particular comments are given below.

General comments:

.) The methodology is relied on Sentinel-2 data from year 2019, and particularly selected monthly intervals. Could you please comment why one year of data and the particular months are selected?

.) Spatio-temporal aggregations are applied on the individual model predictions. This implicitly assumes no temporal change and spatial homogeneity, in this case within one year, and up to 300 m, respectively. Those aspects are partly discussed in the manuscript (the discussion section). Could you please additionally discuss how those aspects affect the minimum mapping unit of your product?

.) 5-fold cross-validation (Section 4.4): Several decisions are made in the validation design, and it is unclear how those settings affects the reported accuracy values:

  • why exactly 5 folds are selected? Does a larger, or smaller number of folds affect the final accuracy values?
  • Are the data split randomly? If yes, please justify why is that preferred here compared to spatially independent folds that offer more objective performance assessment.

.) The test area contains two distinct parts: the flat one and the mountainous one. What is the spatial distribution of the training and validation polygons with respect to those areas? Could you please comment on how different spatial distribution of the test and training data may affect the method performance.

.) Abstract (L32-33) and Conclusions (L482-483): here it is concluded that the methodology is applicable for woody vegetation mapping. This conclusion appears quite extrapolated as it reads as the method is valid globally, but the manuscript demonstrates that the method works on one test site. Please consider to limit this conclusion appropriately.

Particular comments:

L38-39: is it really necessary to have two citation for this statement?

L42: “the detailed forest” - the term is not clear

L48-49: “... remains a challenge” – The statement is too general, please specify why that remains a challenge.

L52: “under cloud-free conditions”- the revisit time should not be affected by clouds. The observation of vegetation is, in the other hand, affected by clouds.

L66: “very high results” - the term is not clear

L75-76: “some classification problems” - the statement is too general - please, specify what problems exactly.

Table 1: This table can be omitted

Figure2: “Tile ID” - is it tile, or granule ID?

L170: “Probability random forest” – could you please comment on why is this method here is in favor of other ML classifiers?

L187-L188: It is unclear here if the spectral info is extracted per pixel or (averaged?) per polygon. From latter text, I assume it is per pixel, but maybe it is good to mention that here, just for the clarity.

Figure 4 and 5: The classes “Quercion frainetto ”, and “Quercion petraea-cerris” that cover about 1/3 of the forest area have smaller accuracy values (ca. 50%). Could you please comment on that?

Figure 4: The MC 3x3, and MC 5x5 for “Pinion mugo”  have much larger accuracy values than other neighborhood-based aggregations. Could you please comment on that?

L282: “largely corresponds” The statement is too general, please specify in which sense the maps corresponds? And how is that concluded (visually, or)?

Section 6.3: This discussion section is nice, but quite logon as well. The authors might considered to shorten it.

L386-417: Those two paragraphs interpret the results; The authors might consider to move them in the results section.

Reviewer 3 Report

This study investigates an approach for woody vegetation classification using temporal and spatial information derived from time series Sentinel-2 images. Random Forest (RF) model was developed and applied to the pixels of each Sentinel-2 images to predict the class probability. The spatial and temporal aggregation were used to determine final woody vegetation classes. The results show that the spatial aggregation of most common class in the neighborhood pixels achieved higher accuracy.

Overall, this is a well-structured manuscript that examined a simple yet interesting classification approach. The authors clearly showed the concept of proposed classification approach. Using a RF probability for spatial and temporal aggregation is a nice idea. Although it is of interest to readers of this journal, I found several issues that would be solved to make the manuscript more understandable.

Major comments:

Although the accuracy assessment of RF model using cross-validation can inform model accuracy, the accuracy metrics are generally biased for mapping entire study area. The accuracy assessment should be based on probability sampling to get unbiased estimator of area and accuracies. There are widely known approaches (e.g. Olofsson et al. 2014. Remote Sensing of Environment 148: 42-57) to cope with this issue. The authors are recommended to consider this kind of accuracy assessment.

The authors resampled Sentinel-2 images to 60 m resolution. What is the motivation for reducing spatial resolution? The original spatial resolution (10 m or 20 m) of Sentinel-2 might reduce the effects of mixed pixels and lead to higher classification accuracy. The authors at least mention the influence of this procedure in the discussion.

Minor comments:

L43: The reference numbers should be placed in one bracket, such as [4–6]. This is the same for other places of this manuscript.

Figure 1: The coordinates, scale, and north arrow are needed.

L151: Is it necessary to divide this into “Methods” and “Methodology” section?

L174: A citation for R software is needed.

L187: In the places training polygon overlaps with multiple Sentinel-2 images, how was the training data extracted? Are there no problems that related to data leakage in time series data?

L191: Why did the authors only choose NDVI? Other spectral indices that use Red-edge bands might improve the accuracy.

L201: I am not sure the reason for including day and month as predictor variables in RF model. I understand that other variables, such as the Day of Year, can indicate the season of image used; however, what kind of influence would be brought by separately using day and month in RF model?

Table 4: OA (%)?

Figure 4 and 5: Although it is easily understandable, the unit (here, %) for y-axis should be provided.

Figure 5. “User’s accuracy”

Table 6: The description of the table is wrong. Please correct the description.

Round 2

Reviewer 2 Report

The authors reacted on my comments from the first review. In addition, a 10-fold validation with random sampling is performed and reported in the manuscript. I do not have any further comments.

Reviewer 3 Report

The authors have responded to all my comments. Now the description of manuscript is clear. I have no additional comments.